



# Monsoons, plumes, and blooms: intraseasonal variability of subsurface primary productivity in the Bay of Bengal

Tamara L Schlosser[1], Andrew J Lucas[1,2], Melissa Omand[3], and J. Thomas Farrar[4]

[1]Scripps Institution of Oceanography, University of California, San Diego, La Jolla, CA, USA.
[2]Dept. of Mechanical and Aerospace Engineering, University of California, San Diego, La Jolla, CA, USA.
[3]Graduate School of Oceanography, University of Rhode Island, Narragansett, RI 02882, USA.
[4]Department of Physical Oceanography, Woods Hole Oceanographic Institution, Woods Hole, MA 02543, USA.

**Correspondence:** Tamara L Schlosser (tamara.schlosser@utas.edu.au)

**Abstract.** During the southwest monsoon, seasonal storms bring torrential rainfall to the South Asian subcontinent and the northern Indian Ocean. Dense cloud cover limits the amount of sunlight that reaches the ocean surface, and sediment-laden river runoff limits the depths to which light can penetrate. Changing light availability should affect phytoplankton primary productivity and its dependent biogeochemical processes, yet little is known about how subtropical weather is linked to ecosystem processes below the ocean's surface. Here, using novel physical and bio-optical measurements from an array of free-drifting, autonomous systems in the Bay of Bengal, we show that the onset of cloudy conditions associated with 'active' monsoon conditions led to >50% reduction in gross chlorophyll productivity (GCP) near the subsurface chlorophyll maximum (SCM) relative to sunny 'break' conditions. Optical backscatter measurements confirm chlorophyll fluorescence fluctuations correspond to biomass variability of a similar scale. Simultaneous bioacoustic measurements collected onboard the autonomous platforms suggest this intraseasonal variability in SCM chlorophyll and biomass generated a response in higher trophic levels. Long-term measurements from biogeochemical (BGC) Argo floats in the bay confirm the presence of intraseasonal oscillations in chlorophyll-*a* concentration with days-to-weeks variability in magnitude similar to the regional annual cycle in the region. Our findings demonstrate that intraseasonal subtropical air-sea variability modulates important regional biogeochemical ocean processes in the Northern Indian Ocean with implications for the Indian Ocean carbon cycle.

## 1 Introduction

The ocean contributes roughly 50% of total global autotrophic carbon fixation, with more than 40% of ocean primary productivity occurring in vast but relatively low biomass oligotrophic seas (Antoine et al., 1996). Our current understanding is that net primary productivity in the subtropical oligotrophic ocean is limited by nutrient supply and not light availability (Mignot et al., 2014), but recently developed autonomous long-duration measurements of subsurface processes allow a more comprehensive analysis of the subsurface variability than has previously been possible (Johnson and Bif, 2021). Less is known about how subtropical primary productivity varies on small time- and space-scales, or its response to the rapidly changing subtropical weather patterns.





Current generation coupled atmosphere-ocean numerical models lack fidelity on 10-90 day timescales in the tropics and subtropics (Gadgil et al., 2005; Mandke et al., 2020; Shroyer et al., 2021). This so-called intraseasonal variability is a major

component of global weather and is fundamental to predicting climate phenomena like the South Asian monsoon (Shroyer et al., 2021). How intraseasonal fluctuations impact biogeochemical processes in the subtropical ocean is poorly understood (Hansell, 2009). Since coupled ocean-atmosphere models struggle to accurately represent the magnitude or the timing of intraseasonal fluctuations (Gadgil et al., 2005; Mandke et al., 2020; Shroyer et al., 2021), these fluctuations must also be inaccurately represented in coupled biogeochemical models. Intraseasonal oscillations like the Madden-Jullian Oscillation

or the Monsoon Intraseasonal Oscillation impact much of the subtropical ocean, so the potential impact on biogeochemical processes may be significant.

Satellite observations reveal the decrease in surface productivity during the southwest monsoon (approximately June to September) in the Bay of Bengal due to the impact of monsoon weather on light availability at monthly scales (Kumar et al., 2010a, b). Light availability at the surface decreases via a reduction in shortwave radiation during the southwest monsoon

(Weller et al., 2016), which oscillate at 10-90 day timescales, categorized as the sunny 'break' period and rainy and cloudy 'active' periods of the monsoon (e.g., Shroyer et al., 2021). Subsurface light availability additionally decreases due to the increasing diffuse attenuation following river run-off, where monsoon rain events inject high concentrations of sediments into the bay, even $\mathcal{O}(100\,\mathrm{km})$ from the coastline (Kumar et al., 2010a). The weather systems that reduce shortwave radiation also impact ocean color satellite measurements, so satellite measurements cannot observe biogeochemical processes over the entire

intraseasonal oscillation. The subsurface impacts of intraseasonal varying light availability are generally unknown, contributing to the uncertainty in our understanding of the Indian Ocean carbon cycle. In particular, stratification, vertical salinity gradients and barrier layers have a dominating influence on subsurface biomass in the Bay of Bengal (Prasanth et al., 2023).

Autonomous technology to measure biogeochemical variability in the ocean – such as the BGC-Argo float program (Argo, 2000) – has provided new opportunities to quantify the variability of the ocean carbon system (Mignot et al., 2014; Johnson

and Bif, 2021). These platforms provide information about biogeochemical transformations below the ocean's surface that are largely invisible to satellite remote sensing (Mis, 2005). Although only recently deployed at scale, autonomous biogeochemical measurements have demonstrated that our capacity to make future predictions of the state of the climate depends sensitively on constraining biogeochemical ocean processes (Gray et al., 2018; Johnson and Bif, 2021).

The BGC-Argo float program typically samples at weekly time scales, which is sufficient to describe the intraseasonal bio-

geochemical variability but not diel processes, such as nonphotochemical quenching (NPQ) and the diel cycle in phytoplankton biomass. Near the surface, NPQ results in a non-biomass-related diel reduction in chlorophyll-$a$ (Chl) fluorescence under high light conditions (Müller et al., 2001). In contrast, the diel cycle is distinct from NPQ in that concentrations of Chl (Marra, 1997; Prasanth et al., 2023), dissolved oxygen (Nicholson et al., 2015; Barone et al., 2019), and particulate carbon (White et al., 2017) increase with increasing irradiance. The diel cycle results from the competing processes of day-time gross pri-

mary production (GPP) and day- and night-time respiration and grazing losses (Marra, 1997). The phase-locking of the diel cycle with irradiance has allowed the development of robust methodologies for estimating GPP and loss rate for a range of biological variables (Marra, 1997; Nicholson et al., 2015). Although the diel cycle directly results from the diel variability in



subsurface irradiance, the sensitivity of the diel cycle to the short-term cloud- or turbidity-driven changes in irradiance remains under-explored.

An improved understanding of the Indian Ocean carbon cycle and its sensitivity to intraseasonal oscillations not only requires a better understanding of primary productivity but also the effect on higher trophic levels. Currently, most autonomous platforms like BGC-Argo deployed long-term do not observe grazers due to the difficulty in making accurate observations, either directly or indirectly. Acoustic backscatter measurements at high frequencies (i.e., 1 MHz) are sensitive to smaller zooplankton species, including grazers, as compared to lower frequency measurements of acoustic backscatter traditionally used to identify

fish and larger marine organisms (Lavery et al., 2007; Ohman et al., 2019; Gastauer et al., 2022). However, without confirming net or video observations (e.g., Lavery et al., 2007), these bioacoustic or high-frequency acoustic backscatter observations (as we present here) remain a relatively uncertain representation of grazer variability.

Here we report on physical, bio-optical, and bio-acoustic measurements gathered in the Bay of Bengal during the 2019 monsoon season, described in the next section. We then quantify the impact of ocean clarity and passing monsoon storms on

subsurface irradiance, and we statistically relate fluctuations in gross production to the subsurface irradiance. We contextualized our observations with an analysis of longer-term BGC-Argo measurements. Finally, we conclude by discussing some of the wider implications of our results, including regional patterns in subsurface productivity we may infer from our results, and evidence of higher trophic level variability. Taken together, our results demonstrate the variability in biogeochemical quantities on intraseasonal time scales, and show how increased autonomous observational capacity can contribute to our understanding

of the processes that shape the ocean climate system.

## 2    Materials and Methods

### 2.1    Field campaign

As part of a campaign to study intraseasonal oscillations in monsoon weather, we deployed an array of three densely instrumented buoys and ocean-wave-powered Wirewalker profiling vehicles (Pinkel et al., 2011) that gathered physical, bio-optical,

and irradiance measurements in July 2019 in the Bay of Bengal. Each Drogued Buoy Air-Sea Interaction System (DBASIS) was deployed for ∼19 days (Fig. 1): M1 on July 9, M2 on July 10, and M3 on July 11, generally following the mesoscale flow. Each profiler had a metocean buoy on the surface, a vertically profiling Wirewalker over the upper 100 m (Pinkel et al., 2011), and six X-wings (1 m$^2$ drag elements) over 200 m to 210 m depth so that the system drifted with the subsurface currents at 200 m. They collected approximately 200 profiles per day spanning the upper 100 m of the water column with sub-meter

vertical resolution.

The buoy's meteorological package measured wind, air temperature, humidity, precipitation, barometric pressure, sea surface temperature, and downwelling solar and infrared radiation. The metocean buoys were equipped with Kipp and Zonen SMP21 shortwave radiation sensors that measure downwelling radiation over wavelengths of 285 nm to 2800 nm (50% points). These measurements were used to estimate the surface photosynthetically available radiation (PAR).



We equipped the Wirewalker profilers with RBR Inc. conductivity, temperature, and depth (CTD) sensors augmented by SBE Wetlabs Ecopucks measuring Chl fluorescence (ChlF, a proxy for Chl biomass), optical backscatter at $532\,\mathrm{nm}$, and chromophoric dissolved organic material (CDOM; not used in the analysis presented here). We measured subsurface downwelling irradiance onboard the Wirewalkers using a Satlantic OCR-504 multi-spectral radiometer at four bands, $380\,\mathrm{nm}$, $412\,\mathrm{nm}$, $490\,\mathrm{nm}$ and $532\,\mathrm{nm}$. All parameters were collected continuously at $6\,\mathrm{Hz}$ and telemetered at that resolution in real-time via RUDICS Iridium modems on the surface buoy. Using the factory calibration, we convert the optical backscatter to Nephelometric Turbidity Unit, $\tau$ (NTU). The median time between profiles was $10\,\mathrm{min}$ at M2 and M3 and $20\,\mathrm{min}$ at M1. For all quantities, we only use measurements during the smooth upward ascent of the profiler.

To estimate the depth-averaged diffuse attenuation, $K_d$ $(\mathrm{m}^{-1})$, the raw measurements of downward irradiance were binned in time and depth to $3\,\mathrm{h}$ and $4\,\mathrm{m}$ bins, respectively, before linearly fitting to an exponential curve. We note fitting a log-transformed curve instead of an exponential curve results in the irradiance fit at all depths being equally weighted, which is problematic when the precision error is not equally weighted (see supplementary materials). We then interpolate back to the original $\mathcal{O}(10\,\mathrm{min})$ time resolution.

We also equipped the profilers with a downward-looking Nortek Signature with a frequency of $1\,\mathrm{MHz}$, from which we show only the acoustic backscatter (Zheng et al., 2022). For each profiler's upcast, we averaged the acoustic amplitudes of each of the four beams over the first 20 bins ($2.5\,\mathrm{m}$ depth range) closest to the transducer. We then interpolated range-averaged beam amplitudes onto a $0.25\,\mathrm{m}$ uniform depth grid before averaging all four beams. The $1\,\mathrm{MHz}$ backscatter was similar to the R/V Sally Ride acoustic backscatter that had a frequency of $150\,\mathrm{kHz}$, with differences between frequencies as expected from literature (Lavery et al., 2007; Ohman et al., 2019; Gastauer et al., 2022).

## 2.2 Chlorophyll-a fluorescence

The primary objective of the field campaign was to investigate air-sea interactions and the underlying physical variability in the BoB. As such, while biogeochemical observations were collected autonomously, we did not sample water in order to field calibrate the chlorophyll sensors. Instead, using laboratory calibration carried out just prior to the experiment, we converted the observed ChlF in relative fluorescence units (RFU) to real units $(\mu\mathrm{g\,L}^{-1})$ via $\mathrm{ChlF} = 0.012 \times (\mathrm{RFU} - 50)$. This conversion has an unknown uncertainty that linearly scales for all values (i.e., percentage error). For context, the observed surface ChlF from the drifting systems and MODIS-Aqua surface Chl from July 2019 were $0.17\,\mu\mathrm{g\,L}^{-1}$ and $0.33\,\mu\mathrm{g\,L}^{-1}$, respectively, with these points located an average of $54\,\mathrm{km}$ apart. The focus of our work is not on the absolute quantities of ChlF but rather on their response to surface and penetrative solar radiation, so we focus on the robust trends in co-variability rather than absolute values.

Chlorophyll pigment concentration, ChlF, and phytoplankton biomass (i.e., carbon content) are highly variable under changing light conditions as phytoplankton adapt to changing light environments by adjusting their light-harvesting pigments (Cullen, 1982, 2015). Due to this, estimates of biomass from backscatter observations are typically a more robust estimate of carbon biomass than ChlF. However, in the situation encountered here, non-algal sources of backscatter, for example in the high-sediment coastal waters we observed at M3 before July 15 (Fig. 1e), backscatter provided an at-times confounded measure




of algal concentrations. Due to this, we compared diel variability in ChlF and backscatter at times and locations where the
backscatter signal was due to algal diel variability. This analysis showed that ChlF was a good proxy at the depth of the sub-
surface chlorophyll maximum, and there we proceed by using ChlF as a proxy for phytoplankton biomass in what follows (see
below).

Near the surface, non-photochemical quenching (NPQ) results in a non-biomass-related diel reduction in fluorescence under
high light conditions (Müller et al., 2001). To correct for NPQ in the ChlF measurements, we follow methods established for
other rapidly profiling platforms (Davis et al., 2008; Todd et al., 2009; Schlosser et al., 2022). We least-squares fit irradiance
at $490\,\mathrm{nm}$ to ChlF over the upper $20\,\mathrm{m}$ over one-day windows, stepping in time every $0.1\,\mathrm{day}$ and averaging overlapping time
steps. We set a maximum correlation coefficient of $-0.1$, ensuring we only modify ChlF if ChlF decreased when irradiance
increased. Although we used only ChlF over the upper $20\,\mathrm{m}$ for the least-squares fitting, we corrected ChlF for all depths with
non-zero irradiance. Corrected ChlF was small with the 95th percentile of $0.016\,\mathrm{\mu g\,L^{-1}}$, and further details are available in the
supplementary materials.

## 2.3 Diel cycle method

After correcting for NPQ, we applied the diel cycle method (Nicholson et al., 2015; Barone et al., 2019) to identify the rate
of gross ChlF production (GCP) and ChlF loss (code available at https://github.com/duebi/dielFit). This loss term describes all
processes resulting in a reduction of ChlF, including grazing, mortality, and downward export. The diel cycle method linearly
fits three terms via ordinary least-squares fitting to the SCM ChlF concentration.

Subsurface ChlF was enhanced at and around the $1021.5\,\mathrm{kg\,m^{-3}}$ isopycnal at all profilers, with this isopycnal having an
average depth of $39\,\mathrm{m}$ at M3 (Fig. 1) that deepened to $50\,\mathrm{m}$ at M1. We hence defined the SCM as the depth-average $1\,\mathrm{m}$ above
and below this isopycnal. Averaging a larger depth range decreased the effectiveness of the diel cycle fit (i.e., the correlation
coefficient increased and mean absolute error decreased). The three fitted terms in the diel cycle method included a constant, a
constant loss rate for both day and night, and a variable in time GCP rate similar to the clear-skies PAR, with day-time growth
maximum at noon and a zero rate of change at night.

Since we collected radiation measurements, we did not need to simulate the solar cycle. We thus modified the code to
optionally include the observed PAR at the SCM to model growth (henceforward referred to as the SCM PAR method). The
SCM PAR method is similar to the sinusoidal growth model but accounts for the impact of cloudiness, changes in diffuse
attenuation, or perturbations of the SCM depth on growth. Before fitting to ChlF, we integrated each growth scheme in time.
For all three profilers and days sampled, we fit each 26-h period (starting at 11 pm local), overlapping each day by one hour.
We additionally modified the diel cycle methodology so that the fitted GCP and loss is always positive.

The performance of the fit was assessed by estimating the correlation coefficient and the mean absolute error, $\mathrm{MAE} = 1/n \sum ||\mathrm{ChlF} - \mathrm{ChlF}_{fit}||$, where $\mathrm{ChlF}$ is the observed ChlF, $\mathrm{ChlF}_{fit}$ is the diel cycle fit, and $n$ is the number of samples. Of
the tested linear, sinusoidal, and SCM PAR schemes for GCP, the SCM PAR method consistently returned the smallest MAE
and better predicted the timing of the observed ChlF maxima by $45\,\mathrm{min}$ on average across all profilers (Table 1). The sinusoid
method also returned a good fit, with a larger MAE, but the linear method was relatively poor in describing the observed ChlF





variability. The fitted mean absolute error doubled when fitting ChlF at a constant depth rather than along-isopycnal due to the vertical heaving of the SCM by passing internal waves (Table 1). The fitting routine was also effective at M1 and M2. At M1,

the timing of peak afternoon ChlF was more variable, particularly when large-amplitude internal waves deepened the SCM during the day (see supplementary materials). We also applied the above diel cycle fitting methodology to turbidity to estimate gross turbidity production (GTP) and turbidity loss. We computed the confidence intervals of the model by bootstrapping the residuals with 200 iterations (Nicholson et al., 2015; Barone et al., 2019).

## 2.4 BGC-Argo analysis

To contextualize our observations and investigate the frequency of light-limited productivity at the SCM in the bay, we analyzed 12 BGC-Argo floats deployed in the BoB, obtained via the OneArgo-Mat routine (Frenzel et al., 2020). These floats were equipped with a CTD and bio-optical sensors (WET Labs ECO-FLBB AP2 or MCOMS, see Johnson et al., 2017) measuring temperature, salinity, pressure, chlorophyll fluorescence (ChlF), optical backscatter coefficient at $700\,\mathrm{nm}$ (at $124°$), as well as other sensors not used here. The data were quality controlled using standard Argo protocols (Wong et al., 2022) and processed

following Uchida et al. (2019). Quality-controlled data was used when available (quality flags 1, 2, 5 or 8), else visual inspection and simple outlier removal was performed (quality flags 0 or 3).

The raw signals were converted to ChlF $(\mu g\,L^{-1})$ and particulate backscattering coefficient $(m^{-1})$ following BGC-Argo procedures (Schmechtig et al., 2018). To account for variations between sensors, we standardized the minimum ChlF and backscatter value by removing the median 'dark' or background value at pressure $>600\,\mathrm{dbar}$ for each float (Uchida et al.,

2019). BGC-Argo floats vary in their temporal and vertical sampling frequencies, so following Uchida et al. (2019) we interpolated the quality-controlled float data onto uniform temporal grids with time steps equal to the minimum temporal sampling rate, which is 10 days for most floats. We standardized the vertical grid from 4 to $1000\,\mathrm{m}$. It had $2\,\mathrm{m}$ resolution in the upper $300\,\mathrm{m}$ and $10\,\mathrm{m}$ resolution below that. This interpolation was performed using a hermite polynomial scheme (pchip in Matlab), but depths outside the observed range were excluded (i.e., no extrapolation). We applied a 3-point moving-median in the

vertical to remove measurement noise and aggregates.

Subsurface ChlF was almost always maximum near the $1022\,\mathrm{kg\,m^{-3}}$ isopycnal, so we defined this isopycnal as the SCM depth. To estimate the SCM ChlF, we average ChlF over the $10\,\mathrm{m}$ above and below this depth. Since the floats profile once every 10 days, and there are too few floats to apply more advanced statistical techniques (Johnson and Bif, 2021), we cannot estimate GCP from diel cycles for the BGC-Argo data. Instead, we examined the floats for any evidence of a correlation

between SCM ChlF and surface and SCM PAR (described below). We note the diel cycle in SCM ChlF is prevalent in the Bay of Bengal (Lucas et al., 2016) and has the potential to alias multi-day trends in ChlF if measurements of ChlF are taken at different local times, as is the case for BGC-Argo measurements. To avoid this aliasing, in regions with diel cycles, ChlF (and other biological or bio-optical variables) should be measured at similar times of the day.





## 2.5 Satellite and reanalysis data

To understand the co-variance of light and BGC-Argo observed ChlF, we considered both MODIS Aqua PAR (NASA Goddard Space Flight Center, 2018b) and the atmospheric reanalysis ECMWF Reanalysis v5 (ERA5) product (Hersbach et al., 2018). On average, the PAR estimated at M3 was less than the satellite derived estimate but similar to the ERA5 estimate. We thereby estimated surface PAR from the ERA5 shortwave radiation, by dividing shortwave radiation by 2.114 (Britton and Dodd, 1976). Then, to estimate PAR at the SCM, we utilized the $490\,\mathrm{nm}$ diffuse attenuation ($K_{d490}$) from the monthly averaged MODIS-

Aqua (Goddard Space Flight Center, 2018a) and converted to a PAR equivalent using the relation in Morel et al. (2007). If this data was missing due to cloud cover, we took the monthly climatology value instead.

## 3 Results

### 3.1 Southwest monsoon campaign

The 2019 field campaign's measurements allowed for a detailed investigation of the co-variability of subsurface physical and

bio-optical properties on an hour-by-hour basis over a nearly 3-week deployment. Conditions at deployment were consistent with the Southwest monsoon 'break' period with weak winds (not shown) and large net heat fluxes ($Q_N$, Fig. 1c). The DBASIS array was deployed at the intersection of two eddies and within a plume with high turbidity and elevated surface chlorophyll-*a* fluorescence (ChlF, Fig. 1d-e). Similar coastal plumes have been found elsewhere at the location of an anticyclonic and cyclonic eddy intersecting (i.e., eddy dipole, Malan et al., 2020). Over 19 days the array drifted southeast, with surface

turbidity and ChlF decreasing after 5 days when the profiler left the coastal plume. Below the surface, the pattern was reversed. At the subsurface chlorophyll maximum (SCM), ChlF was initially relatively low and subsequently doubled after the drifting platforms left the plume (July 14 to 20; Fig. 1d-e).

Diffuse attenuation was strongly influenced by turbidity and was enhanced within the plume, limiting PAR at the SCM ($\pm 1\,\mathrm{m}$ of $1021.5\,\mathrm{kg\,m^{-3}}$ isopycnal, Fig. 2a). After the drifting buoys left the plume, however, turbidity and diffuse attenuation

decreased, leading to a doubling of maximum day-time PAR at the SCM (Fig. 2a), even though surface PAR was relatively steady. This increase in subsurface PAR was coincident with increasing ChlF at the SCM (Fig. 2b). Subsequently, the atmospheric conditions transitioned from mostly sunny conditions to mostly cloudy conditions around July 23 with the onset of the 'active' phase of the Southwest Monsoon (Fig. 1c). Surface-buoy-measured shortwave radiation and net heat fluxes ($Q_N$, Fig. 1c) then decreased and winds increased (not shown). During this time, subsurface PAR dropped significantly and SCM ChlF

decreased, even though near-surface turbidity and $K_d$ were at a minimum (Fig. 1d).

### 3.1.1 Gross ChlF production from diel cycles

The rapid vertical profiling of the Wirewalkers allowed the assessment of bio-optical variability from a time-varying, along-isopycnal frame of reference (Fig. 1d vs. 2b). This approach moderates the effect of passing internal waves that confound measurements taken at discrete depths, and highlighted the variability of ChlF at diel timescales (Fig. 2). The diel cycle at





the SCM showed peak concentrations around 3 hours after local noon and minimum concentrations at dawn (Fig. 2b-c). The time-derivative of SCM ChlF (i.e., $d[\mathrm{ChlF}]/dt$) and SCM PAR (Fig. 2a teal) were both maximum around noon, as has been found previously in the bay (Lucas et al., 2016). All records were corrected for NPQ, which had a fairly small effect on ChlF variability.

The phase-locking of the diel cycle with irradiance has been used to estimate gross production and loss from dissolved oxygen and other biological variables (Marra, 1997; Nicholson et al., 2015; Barone et al., 2019; Johnson and Bif, 2021). Here we adapt these diel models to estimate the gross ChlF production (GCP) and the rate of ChlF loss from the on-isopycnal variability of ChlF and co-located PAR. By modelling growth from the co-located PAR, the impact of variable cloud cover, changes in ocean clarity, or passing internal waves on growth can be accounted for, improving the performance of the model.

The observed diel periodicity and the multi-day trend in SCM ChlF were effectively emulated by the model for each DBA-SIS platform (Fig. 2c and supplementary materials). The fitted rate of ChlF growth was closely coupled to ChlF loss (Fig. 2d), reflecting strong recycling in this oligotrophic setting. When the rate of ChlF loss exceeded GCP, daily-averaged ChlF decreased even when GCP and the diel variability in ChlF remained large (from July 21, Fig. 2c-d).

The observations quantified how variations in GCP closely matched the measured SCM PAR. SCM PAR was a function of shortwave radiation at the surface (largest effect, e.g., July 22 onwards, Fig. 2), diffuse attenuation ($K_d$) in the water column (e.g., July 12 to 15), and the SCM depth (e.g., July 20 to 21 at M1, see supplementary materials). Over the deployment, GCP linearly scaled with the daily averaged shortwave radiation and SCM PAR (Fig. 3a,c) with a significant positive correlation ($r^2 \geq 0.34$, $p < 0.001$, and $n = 40$). From this linear regression, as shortwave radiation decreased by 80% following the transition from the break to active monsoon conditions (July 21 to 26), GCP decreased by 82%. Similarly, as the drifting systems left the coastal plume, SCM PAR and the regression-estimated GCP increased by 58% (July 12 to 21). The fit with in situ subsurface PAR was superior to the fit with shortwave radiation due to the influence of the time-variable diffuse attenuation reflected in that measurement, with days with $K_d^{-1} \leq 15\,\mathrm{m}$ decreasing the performance of the shortwave radiation fit. These findings suggest that the interplay between spatial variability in the magnitude of diffuse attenuation due to ocean clarity and the variability in surface irradiance due to intraseasonal oscillations in monsoon weather act together to influence subsurface productivity in the northern Bay of Bengal.

We also note that due to the SCM depth being approximately $10\,\mathrm{m}$ deeper at M1 compared to the other profilers, SCM PAR was lowest, on average, at this profiler. We can slightly improve the fit (MAE decreased by 12%) of GCP to SCM PAR (Fig. 3b) by accounting for phytoplankton light adaption by relating GCP to PAR/$\overline{\mathrm{PAR}}$ instead of PAR, where $\overline{\mathrm{PAR}}$ is the along-isopycnal average of the SCM PAR at each profiler. This suggests phytoplankton were adapted to a lower irradiance (i.e., isolume) at M1 than the other profilers, but regardless, GCP similarly increased for larger SCM PAR.

Despite the occasional non-algal sources of high turbidity, especially at M3, we estimated a good diel cycle fit ($r^2 \geq 0.4$) in turbidity for 28 days in total across the three profilers. The estimated gross turbidity production (GTP) linearly varied with surface and SCM PAR, similar to GCP variability (Fig. 3b,d). This indicates the co-variability of ChlF and light availability is not limited to pigment concentration (e.g., photo-acclimation) but also corresponds to changes in biomass.



## 3.2 Barrier layer stability

In the southern BoB, SCM productivity was linked to barrier layer thickness, defined as the depth-region between the MLD base and the isothermal layer (ITL) (Prasanth et al., 2023). Here, we define the ITL depth where temperature is $\leq 1\,^{\circ}\mathrm{C}$ the sea surface temperature, and the MLD as the depth where the density increase from the surface value corresponds to a temperature increase of $1\,^{\circ}\mathrm{C}$. The ITL was on-average within $1.5\,\mathrm{m}$ of the SCM depth at M3 and the barrier layer thickness was highly dependent on salinity variations (Fig. 4a-c).

While M3 was within the coastal plume (before July 15), the mixed layer depth (MLD) approximately equalled the SCM depth, which continued til just before July 17.8 (Fig. 4a-b). While within the plume, water temperature and salinity at the surface and MLD were different, indicating a very thin barrier layer was present, but ChlF was the same (Fig. 4d-f). A passing storm resulted in a shallow rain-layer forming around July 17, the rapid shoaling of the MLD, diverging surface and MLD temperature and salinity, and a rapid decrease in ChlF at the MLD base (Fig. 4). Over the remaining observation period,

the MLD deepened until it reached the SCM around July 25. ChlF at the MLD and SCM then converged, and surface ChlF increased despite the transition to active monsoon conditions and decrease in light availability. This may indicate that the erosion of the barrier layer contributed to the observed decrease in ChlF and GCP upon the transition to active monsoon conditions (Fig. 2d). The high SCM productivity and GCP during the sunny break conditions around July 20 were hence likely supported by the thick barrier layer that formed from the rain event, isolating upper ocean mixing from the SCM.

## 270 3.3 Regional and seasonal context

To determine whether our observations of co-varying SCM PAR and GCP in the central Bay of Bengal were representative of the region and season, we analyzed the BGC-Argo subsurface ChlF, ERA5 estimated PAR and MODIS-Aqua diffuse attenuation climatology (see Sections 2.4 and 2.5). To illustrate, we show one example of positively correlated SCM PAR and ChlF over 12 months (WMO ID: 2902193; $r^2 = 0.58$, $p < 0.001$, and $n = 70$; Fig. 5). At the seasonal time scale, trends in SCM PAR

were more similar to SCM depth than the incident surface PAR, where PAR exponentially decays with depth and so can largely control subsurface PAR. In contrast, during the monsoon months of June to September, temporal variability was comparable between surface PAR and SCM ChlF. This intraseasonal signal was sufficiently strong such that during the southwest monsoon period, surface PAR and subsurface ChlF (averaged over the upper $140\,\mathrm{m}$) were significantly positively correlated ($r^2 = 0.26$, $p < 0.01$, and $n = 27$), even without accounting for changes in SCM depth.

The BGC-Argo float observed the largest change in ChlF at the SCM during the southwest monsoon between the end of July and start of August. The minimum and maximum PAR estimated at the SCM around this time were on July 23 and August 6, with observed ChlF on these days of $0.38\,\mathrm{\mu g.L^{-1}}$ and $1.65\,\mathrm{\mu g.L^{-1}}$, respectively (Fig. 5). The time of observation will also impact ChlF in this region of strong diel cycles. We note the time of sampling of the minimum and maximum SCM PAR was at 4.12 pm and 3.08 am local, respectively, where the DBASIS float observed daily maximum ChlF at 3 pm and minimum at 7 am,

on average (Fig. 2c). The over 300% increase in ChlF at the SCM observed by BGC-Argo was not only due to the transition from active to break monsoon conditions (Fig. 5), but also the shoaling of the SCM that further increased light availability.





If we contrast to the DBASIS platform observations, at M3 ChlF decreased from $0.96\,\mu g.L^{-1}$ to $0.58\,\mu g.L^{-1}$ on July 21 to 25, and at the same local time as the minimum and maximum BGC-Argo sample (Fig. 2c). This equates to a 40% percent difference at M3, compared to the 77% difference in the BGC-Argo float. We note the change in gross ChlF production at the DBASIS platform was double the ChlF change, at an $\geq$80% decrease, where this number is independent of sampling time (Fig. 2e). This may suggest changes in gross ChlF production due to intraseasonal oscillations at the BGC-Argo exceeded what we observed at M3, given variability in ChlF was larger at the BGC-Argo float than at M3.

## 4 Discussion

### 4.1 Primary productivity during the southwest monsoon

During the 2019 monsoon in the central Bay of Bengal, we observed a diel periodicity in both ChlF and turbidity at the subsurface chlorophyll maximum (SCM, Figs. 1 and 2). This diel periodicity has been observed elsewhere in the BoB (Lucas et al., 2016; Prasanth et al., 2023), and highlights a tight coupling between phytoplankton growth and loss in these oligotrophic waters. As both the ChlF and turbidity derived gross production exhibited a diel pattern and varied with PAR, we conclude the SCM represents a maximum in biomass, as has been previously shown in the Indian Ocean waters (Cornec et al., 2021; Prasanth et al., 2023). Estimated gross ChlF production (GCP) at the SCM responded to variations in light from passing monsoon storms and changing ocean clarity (Fig. 3b and supplementary materials). GCP estimated from the diel cycle linearly scaled with surface shortwave radiation at each DBASIS system but the fit was sensitive to changing ocean color and turbidity. By accounting for variations in diffuse attenuation by linearly scaling GCP with PAR at the SCM rather than at the surface, we improved the linear fit (Fig. 3). We conclude we observed light limited SCM growth, contrary to prior studies showing the subtropical oligotrophic waters were typically nutrient limited (e.g., Mignot et al., 2014).

The estimated SCM GCP rapidly decreased following the transition from sunny 'break' to cloudy 'active' monsoon conditions by $\geq$80% in the northern BoB (Fig. 2d-e). This intraseasonal signal in ChlF gross production was also evident in BGC-Argo ChlF measurements from central BoB, with a similar difference in subsurface ChlF levels during the active vs. break periods as that observed by our measurements (e.g., July 24 to August 8 in Fig. 5a). Due to the southwest monsoon, surface PAR fluctuates on timescales of days to months over the entire bay (Gadgil et al., 2005; Mandke et al., 2020; Shroyer et al., 2021). Average surface PAR increased from 20°N to 8°N, while the monthly standard deviation in surface PAR decreases from north to south (Fig. 6). This indicates that southern BoB waters have higher and more consistent surface PAR, and towards the north, intraseasonal oscillations have a larger impact on surface PAR. We hence predict our observed $\geq$80% difference in SCM productivity due to intraseasonal oscillations was representative of northern and perhaps central BoB variability during the southwest monsoon, confirmed by the BGC-Argo observations from central BoB (Fig. 5). From the surface PAR fluctuations, we expect weaker but still prevalent oscillations in productivity to occur in more southern waters.

Observations from the southern BoB have highlighted the importance of the barrier layer in supporting SCM productivity by impeding the mixing of the SCM into the mixed layer (Prasanth et al., 2023). Our observations deviated from the southern BoB observation but still confirmed the importance of the barrier layer. The formation of a shallow rain-layer rapidly shoaled



the mixed layer, increasing the barrier layer thickness, and isolating the SCM from being eroded by upper ocean mixing (Fig. 4). Had the sunny break conditions not followed after a storm event, we expect observed intraseasonal oscillations in ChlF and GCP would have decreased. At M3, the erosion of the barrier layer co-occurred with minimum SCM ChlF rather than when ChlF was enhanced, like observations by Prasanth et al. (2023). The erosion of the barrier hence had minimal impact on the already decreased SCM productivity, but likely contributed to the observed decrease in ChlF. We also note that at M3 the bulk

of the SCM productivity was below the barrier layer, while Prasanth et al. (2023) observed enhanced ChlF within the barrier layer when it was present. As the SCM is typically located at the deepest depth where PAR is sufficient for growth and the shallowest depth where nutrients are available, our observations may suggest a relatively deeper nutricline than in the southern BoB. The barrier layer still separated the SCM from upper ocean mixing, and hence will influence SCM productivity, as found in the southern BoB (Prasanth et al., 2023).

Both the DBASIS and BGC-Argo observations indicate the intraseasonal variability in surface light project onto the SCM primary productivity. These observations suggest propagating coupled ocean-atmosphere intraseasonal weather patterns may leave a biological footprint of time- and space-variable primary productivity within the Bay of Bengal. Barrier layer formation is additionally impacted by intraseasonal oscillations (Thadathil et al., 2007; Girishkumar et al., 2011; Prasanth et al., 2023), and will impact ChlF vertical distributions (Prasanth et al., 2023) and hence SCM productivity. Other biogeochemical properties

may also be dependent on intraseasonal weather patterns, since primary productivity modulates dissolved oxygen, particulate organic carbon, and dissolved inorganic carbon concentrations (Sarma et al., 2012). Similar patterns could also be evident in other tropical and subtropical oceans subject to intraseasonal atmospheric variability.

## 4.2 Higher trophic levels

Although we focus on the light limitation on primary productivity in this manuscript, grazing by higher trophic levels can
also constrain phytoplankton variability. To provide some information on this component of the food web, we used acoustic backscatter measurements from 1 MHz Acoustic Doppler Current Profilers onboard the Wirewalkers (Fig. 7, Ohman et al., 2019). The backscatter measurements showed a shallow diel migration of zooplankton between around 60 m at daytime to near the SCM at nighttime (Fig. 7). Although we do not have observations to confirm the size and species present, as highlighted in the 'Introduction', previous analysis with a 1 MHz acoustic backscatter confirms its sensitivity to smaller zooplankton species,
including grazers (Lavery et al., 2007; Ohman et al., 2019; Gastauer et al., 2022). These smaller species tend to have slower and shallower vertical migrations than larger species (Gastauer et al., 2022), so the shallow diel migration in Fig. 7b is consistent with a small zooplankton species.

The acoustic backscatter strength near the SCM increased over time and reached a maximum on July 24 (Fig. 1c and 7b). During this period of increasing acoustic backscatter, the rate of ChlF loss began to exceed that of GCP (Fig. 2d), leading to a
decrease in ChlF from July 21. This suggests that grazing could have contributed to the increased loss rate and might provide a link between intraseasonal variability in primary production and higher trophic levels. Prasanth et al. (2023) highlighted that a better understanding of barrier layer formation under variable phytoplankton growth rates and zooplankton grazing is required.





Here, we reveal important linkages between these processes and additionally highlight the importance of the intraseasonal oscillations in imparting periodicity to phytoplankton growth at the SCM.

**5 Conclusions**

The coupled bio-optical and physical observations collected from the DBASIS drifting buoy-profiler systems allowed concurrent measurements of the subsurface irradiance and the diel cycles of subsurface chlorophyll fluorescence during the southwest monsoon (Fig. 2b). These observations showed that changes in subsurface irradiance in the northern Bay of Bengal were coupled with intraseasonal variability in monsoon weather and spatial variability in the turbidity of surface waters, and led to major

changes in subsurface primary productivity.

Coupled ocean-atmosphere global forecasting models are challenged to accurately represent intraseasonal variability in the tropical and subtropical ocean. Here we have shown that intraseasonal variability extends to the subsurface biogeochemical properties of the Northern Indian Ocean. Representing this variability correctly in climate predictions is necessary to reduce forecast uncertainty since it impacts both the ocean carbon system and the optical characteristics of the upper ocean, which

modulate ocean heat content. Continued advances in autonomous measurement techniques, and widespread deployment of those platforms, is necessary to improve our understanding of the contribution of intraseasonal biogeochemical variability to the ocean ecosystem.

*Code and data availability.* Data underlying the presented figures and tables are provided on reasonable request from the corresponding author till 2023. Data will be accessible without request after this period. Biogeochemical-Argo data are freely available through one of the

two Global Data Assembly Centers (GDAC), using the WMO number of the float, which is its specific identifier. Argo data were collected and made freely available by the International Argo Program and the national programs that contribute to it. (http://www.argo.ucsd.edu,http://argo.jcommops.org). The Argo Program is part of the Global Ocean Observing System.

Ocean color satellite measurements are freely available through the NASA Goddard Space Flight Center, Ocean Ecology Laboratory, Ocean Biology Processing Group (2018). doi: 10.5067/AQUA/MODIS/L3M/KD490/2018, 10.5067/AQUA/MODIS/L3M/PAR/2018, and

10.5067/AQUA/MODIS/L3M/CHL/2018. Accessed August 2020 to October 2021. The VGPM NPP measurements were provided by the Ocean Productivity team at Oregon State University (http://sites.science.oregonstate.edu/ocean.productivity/index.php). The altimeter products were produced and distributed by the EU Copernicus Marine Service Information (doi: 10.48670/moi-00021). The modified diel cycle code will be publicly available as a version update of the currently available code (https://github.com/duebi/dielFit). All other code used is available upon request.

*Author contributions.* The manuscript contains observations from the Office of Naval Research MISO-BOB 2019 field campaign. AJL and JTF led the conception, planning, and execution of the field experiment and TLS assisted in the execution. TLS conducted the majority of



the data analysis and MO contributed the biogeochemical Argo analysis. TLS and AJL drafted the initial manuscript. All authors approved the final manuscript.

*Competing interests.* The authors declare that they have no conflict of interest.

*Acknowledgements.* We thank the crew, volunteers, and scientists who aided in the field data collection aboard the R/V Sally Ride. Finally, we would like to thank Professor Debasis Sengupta of the Indian Institute of Science, Bangalore for his contributions to our understanding of intraseasonal variability in the Northern Indian Ocean. TLS, AJL, and JTF were supported by the Office of Naval Research "Monsoon Intra-Seasonal Oscillations Bay Of Bengal" (MISO-BOB) program. TLS and AJL were supported by ONR N00014-17-1-2391 and AJL was supported by ONR N00014-17-1-2987. MO was supported by AWD05945 and NSF 2048491.





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





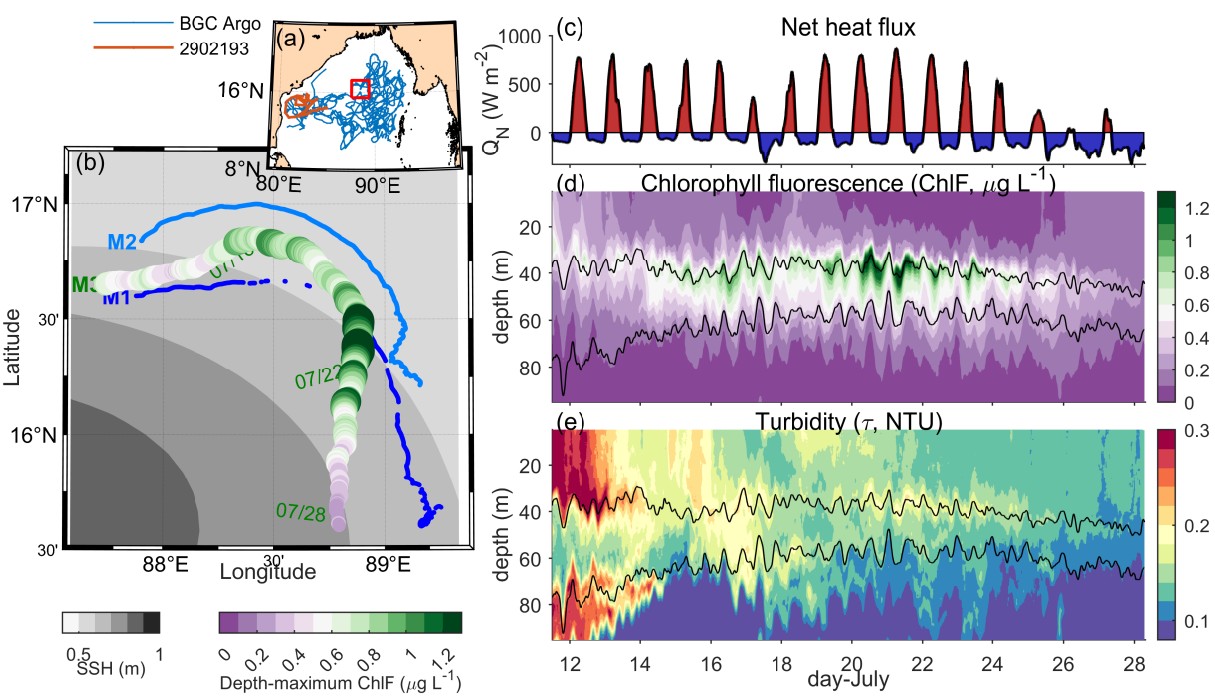

**Figure 1.** Overview of observations. a) Bay of Bengal with the study region (red square), location of analyzed biogeochemical (BGC) Argo profilers deployed from 2013 to 2018 (blue, WMO ID: 2902086, 2902087, 2902114, 2902158, 2902160, 2902161, 2902189, 2902196, 2902217, 2902264), with a further analyzed 2016 profile highlighted (orange, WMO ID:2902193, Argo, 2000). b) Position and date (month/day) for the three high-resolution profilers: M1 (dark blue), M2 (light blue), and M3, with the depth-maximum chlorophyll-*a* fluorescence (ChlF) shown for M3. In grey, we contour the EU Copernicus Marine Service global ocean $1/12°$ sea surface height (SSH) from the 4th of July (https://doi.org/10.48670/moi-00021). At M3, c) the net heat flux ($Q_N$), d) ChlF, and e) turbidity ($\tau$), with two isopycnals ($1021.5\,\mathrm{kg\,m^{-3}}$ and $1023\,\mathrm{kg\,m^{-3}}$) contoured (black) in panels d) and e).



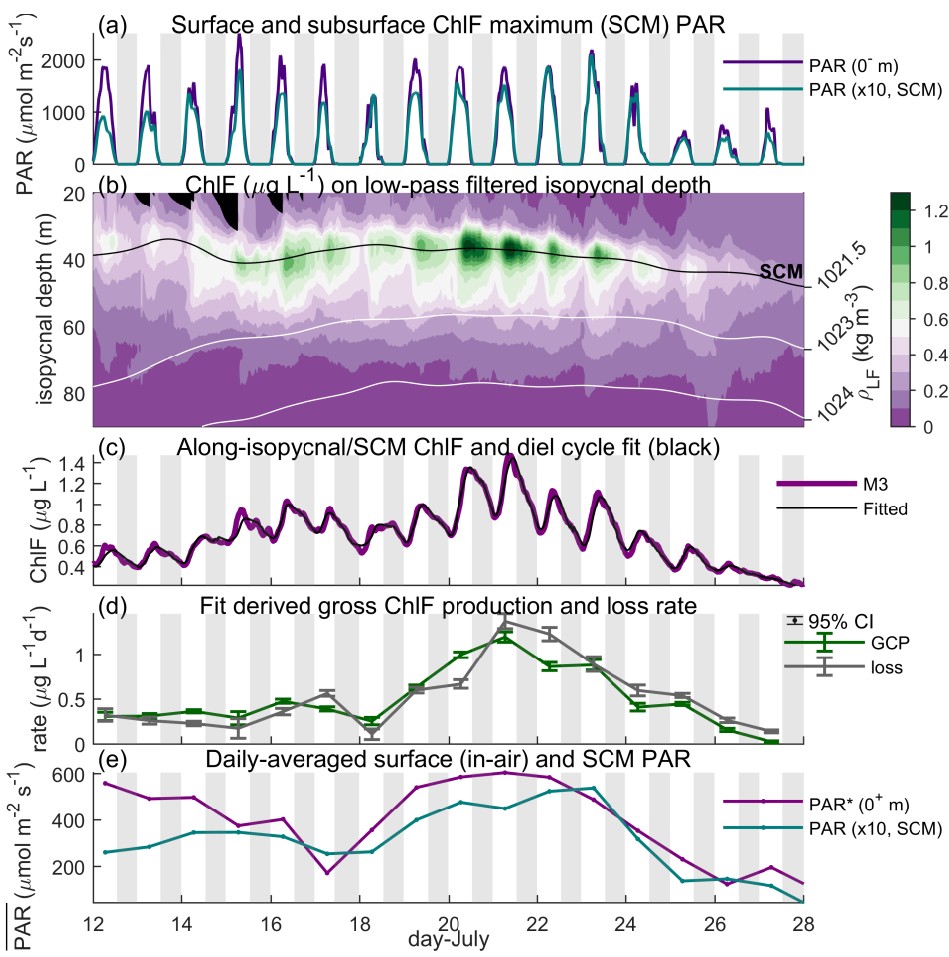

**Figure 2.** Co-varying light and chlorophyll. At M3, a) photosynthetically available radiation (PAR) at the surface (purple) and subsurface chlorophyll maximum (SCM, teal) multiplied by 10. b) On isopycnal ChlF, with depth representing the 48-h low-pass filtered isopycnal depth (white contour, right y-axis) and the SCM depth labelled (black contour). c) SCM ChlF (purple) and the diel cycle fit (black, see "Methods"). d) Rate of gross ChlF production (GCP, green) and loss (grey), with the 95% confidence interval labeled (error bars). e) similar to panel a) but daily averaged (denoted by the over-bar). Note $0^+$ and $0^-$ m indicate surface measurements in air and water, respectively, and we denote a conversion from shortwave radiation to PAR with $(\cdot)^*$.



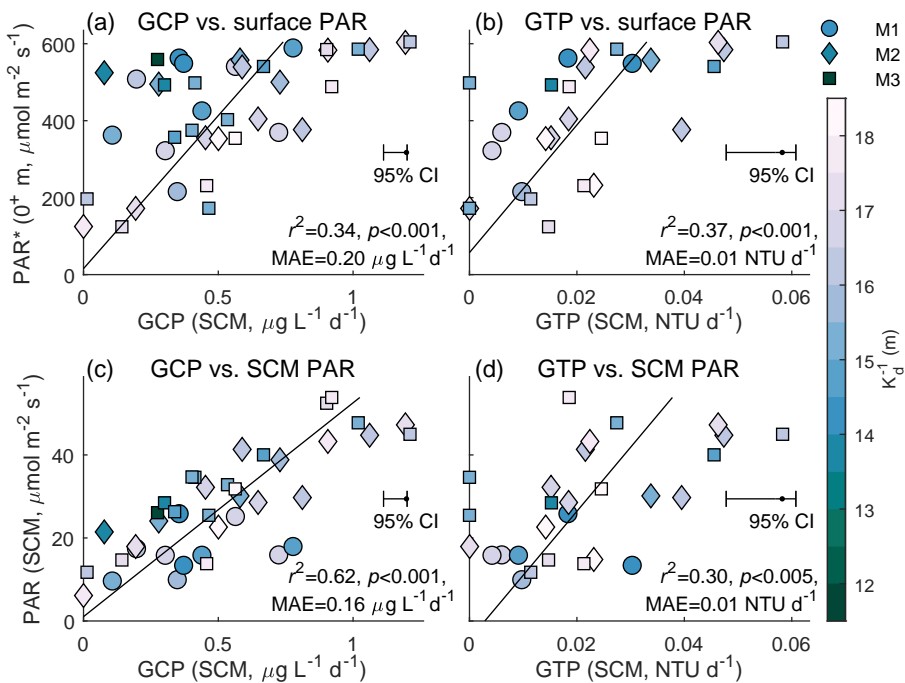

**Figure 3.** Light-limited productivity. Daily averaged surface PAR (y-axis, converted from shortwave radiation) vs. a) GCP and b) gross turbidity production (GTP) for all profilers and sample days with a good diel cycle fit ($r^2 \geq 0.4$), colored corresponding to the inverse diffuse attenuation ($K_d^{-1}$). b,d) Similar to the above panel, but with PAR from the SCM. We least-squares linear fit each panels gross production and light variable, labeling the resulting linear regression (black), $r^2$, $p-$value, and mean absolute error (MAE).



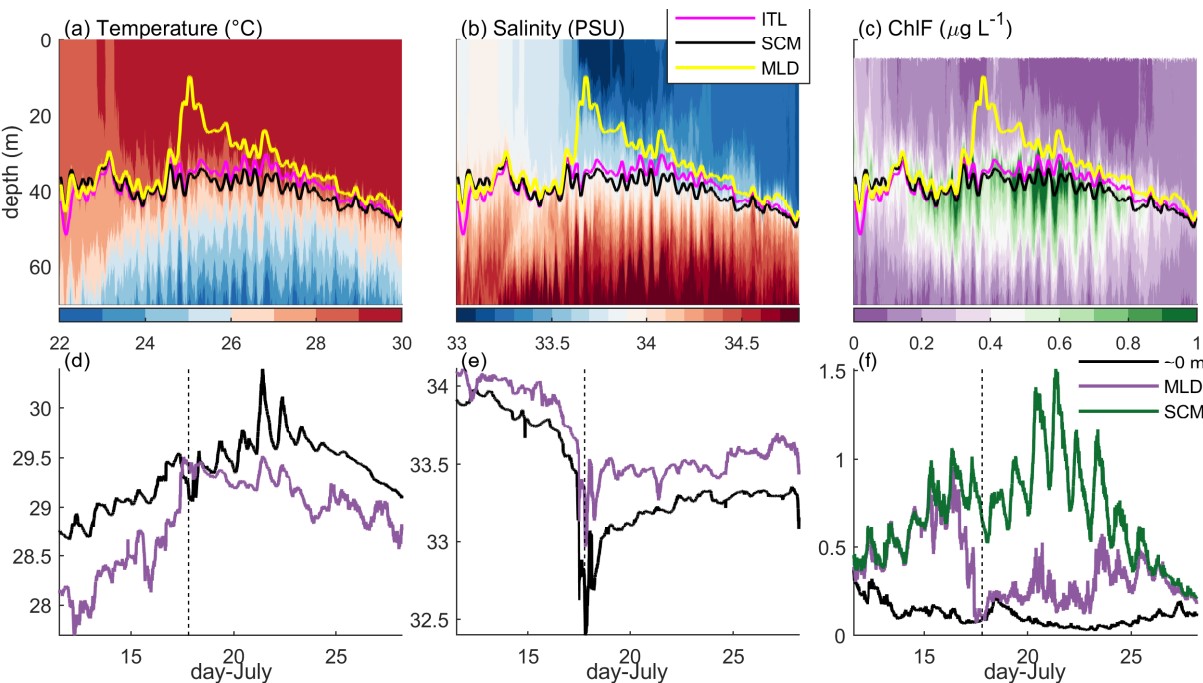

**Figure 4.** Barrier layer vs. the SCM. At M3, a,d) temperature, b,e) salinity, and c,f) ChlF, over the upper 70 m (top panels) and at specific depths (bottom panels). a-c) Depth of the isothermal layer (ITL, pink), SCM (black), and mixed layer depth (MLD, yellow). d-f) Variability at near-surface (black), and mixed layer depth (purple). f) Variability at the SCM (green).





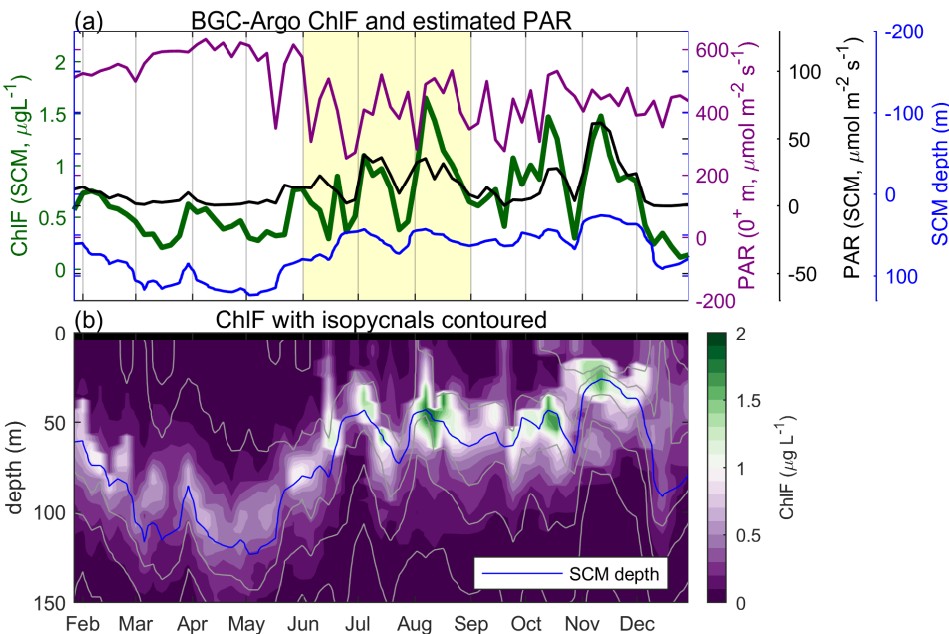

**Figure 5.** Co-varying light and ChlF from a 2016 BGC-Argo deployment (WMO ID: 2902193, Figure 1a). ChlF concentrations
a) depth-averaged over the $20\,\mathrm{m}$ centered on the $1022\,\mathrm{kg\,m^{-3}}$ isopycnal we designate as the SCM (green, left y-axis) and b) contoured over
depth. Also on panel a), for each BGC-Argo profile we find the closest ERA5 surface PAR (purple, right y-axis, Hersbach et al., 2018) and
MODIS-Aqua $490\,\mathrm{nm}$ diffuse attenuation (Goddard Space Flight Center, 2018a), converted a PAR equivalent (Morel et al., 2007), to
estimate SCM PAR (black, right y-axis). Yellow shading in panel a) indicates the southwest monsoon period. In panel b), isopycnals are
contoured in gray every $1\,\mathrm{kg\,m^{-3}}$, with the SCM plotted in blue.





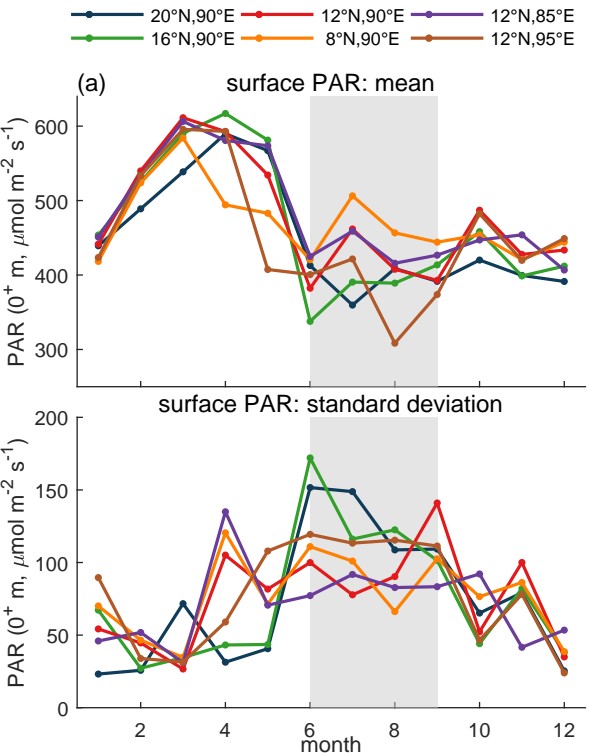

**Figure 6.** ERA5 derived monthly climatology (2012-2022) of surface PAR. ERA5 PAR at fixed locations around the bay (color), a) monthly averaged and b) monthly standard deviation. Gray shading indicates the southwest summer monsoon period.



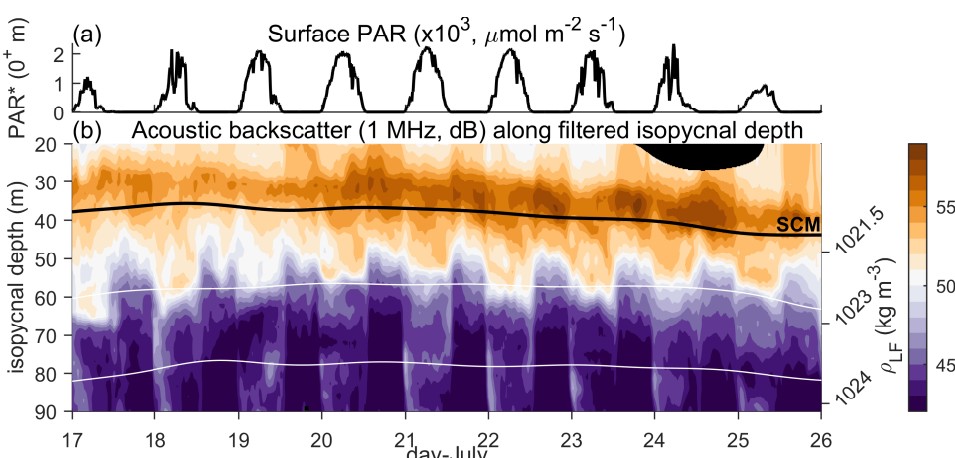

**Figure 7.** Diel vertical migration to the SCM. a) Surface PAR (converted from shortwave radiation), and b) on isopycnal acoustic backscatter at 1 MHz (depth as Figure 2b).





**Table 1.** The performance of the diel cycle fit averaged over all three profilers and days fitted. We tested three methods of estimating the gross chlorophyll production (GCP) along an isopycnal ($1021.5\,\mathrm{kg\,m^{-3}}$) estimated as the subsurface chlorophyll maximum (SCM): linear, sinusoidal and from the observed SCM PAR. We also applied this method at a fixed depth corresponding to the average depth of the SCM at each profiler, labelled $\overline{\mathrm{SCM}}$ PAR where ($\bar{\cdot}$) indicates an average in time. We show the estimated correlation coefficient ($r^2$) and mean absolute error (MAE) from fitting ChlF, and we estimate the MAE in the timing of maximum ChlF ($\mathrm{MAE}_t$).

| Method | $r^2$ ($p$-value) | MAE ($\mathrm{\mu g\,L^{-1}}$) | $\mathrm{MAE}_t$ (h) |
|---|---|---|---|
| Linear | $0.69\ (3.7 \times 10^{-4})$ | 0.042 | 3.5 |
| Sinusoidal | $0.78\ (6.7 \times 10^{-4})$ | 0.032 | 2.1 |
| SCM PAR | $0.81\ (6.7 \times 10^{-6})$ | 0.029 | 1.4 |
| $\overline{\mathrm{SCM}}$ PAR | $0.49\ (1.0 \times 10^{-2})$ | 0.060 | 2.2 |