# Peer review of "Monsoons, plumes, and blooms: intraseasonal variability of subsurface primary productivity in the Bay of Bengal"

_EGUsphere, 2025_

## Referee Comment (RC3)

Overall, this is a well-written paper with solid analysis and provides some useful new insights into the role of intraseasonal oscillations in driving variability in chlorophyll fluorescence and gross chlorophyll productivity at the subsurface chlorophyll maximum. I have found no major issues with the analysis or interpretation, but I have highlighted some minor revisions below.

L35 – should "oscillate" be "oscillates"?

L99-101. "We note fitting a log-transformed curve..." – I don't understand this sentence. You didn't fit a log-transformed curve, you fitted an exponential curve. You also refer to supplementary materials, but I don't see any discussion of the precision error being "not equally weighted" in these supplementary materials. Please clarify this sentence.

Supplementary material and units of I. In the supplementary materials, you state that I is the vertical change in irradiance, with units  $\mu W \ cm^{-2} \ nm^{-1}$ . Firstly, it's not clear why you use  $\mu W \ cm^{-2}$  rather than  $W \ m^{-2}$ ? A conversion factor would be involved, but the numbers would only be different by  $10^{-2}$ . Secondly, if this is a *vertical change* in irradiance then shouldn't these units be divided by vertical distance (e.g., per metre)?

L134: "Corrected ChlF was small with the 9th percentile of 0.016 [micrograms per L]" – do you mean that the **correction** to ChlF was small? Figure S2 seems to support this, but actual values of ChlF were much larger (order 1).

L181 – Why is the SCM at a deeper/denser isopycnal in the BGC argo float data than the wirewalker observations?

L202: "deployed at the intersection of two eddies" – is this shown anywhere? How do you know this to be the case?

Fig. 2. I find it very hard to follow which day is which from the bottom panel to the top panel. I think this figure would be improved if panels (a) and (b) were switched, so that the daily-averaged values in panel (e) could be more easily linked to the raw values in (a).

In addition, Fig. 2 has numerous instances of poor formatting – in particular, the title text for several panels overlaps with the plotting area. The colorbar for panel (b) is also not labelled.

L221: "The time-derivative of SCM ChlF" – is this shown anywhere? Perhaps a better way to phrase this would be to say that PAR is maximum at this time and the SCM ChlF typically increases rapidly, as seen in Fig. 2c.

L239-244: For a while, I was confused by this paragraph as I thought subsurface PAR was PAR at -0 m (I know you refer to the latter as surface PAR in water, but hopefully you can see where the confusion comes from). I would suggest using SCM PAR rather than "subsurface", especially here.

L256: "temperature <= 1°C the sea surface temperature" – insert "lower than" or similar.

L257: "density increase... corresponds to a temperature increase of 1 °C" – I think you mean decrease? Increasing temperature decreases density!

L261: "July 17.8" – typo?

L264-269: I'm not convinced by this argument, I think it needs more evidence. From what I can see, the SCM ChlF is not strongly affected by the shoaling of the MLD. There is almost no effect of the MLD shoaling from day 16-18 on SCM ChlF – this only increases after 20 July as the MLD increases in depth. Therefore, the final two sentences of this paragraph seem very speculative.

L315: "from southern BoB" -> "from the southern BoB"

---

## Author Comment (AC2)

**Overall, this is a well-written paper with solid analysis and provides some useful new insights into the role of intraseasonal oscillations in driving variability in chlorophyll fluorescence and gross chlorophyll productivity at the subsurface chlorophyll maximum. I have found no major issues with the analysis or interpretation, but I have highlighted some minor revisions below.**

We thank the reviewer for their constructive feedback. We have carefully reviewed all comments and improved the manuscript accordingly. We have copied the reviewer comments in bold and responded below them.

**L35 – should "oscillate" be "oscillates"?**

Adjusted.

**L99-101. "We note fitting a log-transformed curve…" – I don't understand this sentence. You didn't fit a log-transformed curve, you fitted an exponential curve. You also refer to supplementary materials, but I don't see any discussion of the precision error being "not equally weighted" in these supplementary materials. Please clarify this sentence.**

We have simplified the text to: "To estimate the diffuse attenuation averaged by depth, Kd (m−1), the raw measurements of downward irradiance were grouped in time and depth into 3 h and 4 m bins, respectively, and then least-squares fitted to an exponential curve. Irradiance was then interpolated back to the original O(10 min) time resolution (see supplementary materials for further details)."

**Supplementary material and units of $I$. In the supplementary materials, you state that $I$ is the vertical change in irradiance, with units $\mu W\ cm^{-2}\ nm^{-1}$. Firstly, it's not clear why you use $\mu W\ cm^{-2}$ rather than $W\ m^{-2}$? A conversion factor would be involved, but the numbers would only be different by $10^{-2}$. Secondly, if this is a *vertical change* in irradiance then shouldn't these units be divided by vertical distance (e.g., per metre)?**

We have modified the supplementary materials so we use scientific notation. We clarify that irradiance should be the irradiance over depth, not vertical change in irradiance. In the supplementary, we change "vertical change in irradiance" to "depth-variable irradiance".

**L134: "Corrected ChlF was small with the 9th percentile of 0.016 [micrograms per L]" – do you mean that the correction to ChlF was small? Figure S2 seems to support this, but actual values of ChlF were much larger (order 1).**

Yes, our text is incorrect. We have fixed to "The NPQ correction to ChlF was small, with a 95th percentile of 0.016 µg L−1. Further details are available in the supplementary materials." Thank you for identifying this.

**L181 – Why is the SCM at a deeper/denser isopycnal in the BGC argo float data than the wirewalker observations?**

The difference in isopycnal may be due to the difference in record length between the Wirewalker (19 days) and the BGC-Argo float (~1 year), or the sampling frequency of these profilers. For example, the BGC-Argo observed a bloom between 1020 kg m-3 and 1023 kg m-3 in November, and if it had only sampled this bloom, we may not have selected the 1022 kg m-3

contour as the SCM but a slightly less dense contour. We have added to the text, "Individual months could have enhanced ChlF near a slightly different isopycnal, like the 1021.5 kg m−3 isopycnal used when analyzing the DBASIS floats, but we simply use the same isopycnal year-round."

**L202: "deployed at the intersection of two eddies" – is this shown anywhere? How do you know this to be the case?**

We have replaced Fig. 1 with two new figures. The new Fig. 1 shows satellite Chl, the coastal plume, and the two eddies. We now state on lines 220-225, "The DBASIS array was deployed at the intersection of two eddies, as inferred from sea surface height (SSH, Fig. 1). Low-SSH (SSH < 0.5 m) and high-SSH (SSH > 0.5 m) eddies were present to the northwest and southwest, respectively. Between these two eddies, a plume of elevated surface chlorophyll was advected more than 400 km from the coast toward the DBASIS array (Fig. 1a). This coastal plume exhibited elevated turbidity and surface chlorophyll-a fluorescence (ChlF), both of which reduced light penetration into the upper ocean (Fig. 2). Similar coastal plumes have been observed elsewhere at the intersection of cyclonic and anticyclonic eddies (i.e., eddy dipole, Malan et al., 2020)."

**Fig. 2. I find it very hard to follow which day is which from the bottom panel to the top panel. I think this figure would be improved if panels (a) and (b) were switched, so that the daily-averaged values in panel (e) could be more easily linked to the raw values in (a).**

**In addition, Fig. 2 has numerous instances of poor formatting – in particular, the title text for several panels overlaps with the plotting area. The colorbar for panel (b) is also not labelled.**

We have made these adjustments and updated figure citations accordingly.

**L221: "The time-derivative of SCM ChlF" – is this shown anywhere? Perhaps a better way to phrase this would be to say that PAR is maximum at this time and the SCM ChlF typically increases rapidly, as seen in Fig. 2c.**

We have adjusted the text to, "The noon maximum in $PAR_{SCM}$ (Fig. 3b teal) coincided with a rapid increase in $ChlF_{SCM}$, as found previously in the bay (Lucas et al., 2016)".

**L239-244: For a while, I was confused by this paragraph as I thought subsurface PAR was PAR at -0 m (I know you refer to the latter as surface PAR in water, but hopefully you can see where the confusion comes from). I would suggest using SCM PAR rather than "subsurface", especially here.**

We have changed 'subsurface PAR' to '$PAR_{SCM}$' here and elsewhere in the manuscript. We also clarified the first and last sentence of this paragraph to improve readability.

**L256: "temperature <= 1°C the sea surface temperature" – insert "lower than" or similar.**

Adjusted.

**L257: "density increase… corresponds to a temperature increase of 1 °C" – I think you mean decrease? Increasing temperature decreases density!**

Yes, thank you for identifying our error.

**L261: "July 17.8" – typo?**

Fixed.

**L264-269: I'm not convinced by this argument, I think it needs more evidence. From what I can see, the SCM ChlF is not strongly affected by the shoaling of the MLD. There is almost no effect of the MLD shoaling from day 16-18 on SCM ChlF – this only increases after 20 July as the MLD increases in depth. Therefore, the final two sentences of this paragraph seem very speculative.**

We have modified the text to, "The thick barrier layer that formed from the rain event should theoretically isolate upper ocean mixing from the SCM. The observed increase in $ChlF_{SCM}$ and $GCP_{SCM}$ from July 18 may therefore be due to both the sustained sunny break conditions and a decrease in SCM mixing from the barrier layer formation (Fig. 3d)."

**L315: "from southern BoB" –> "from the southern BoB"**

Adjusted.

---

## Author Comment (AC3)

**RC2: 'Comment on egusphere-2025-4206', Anonymous Referee #2, 05 Nov 2025**

**This study provides valuable insights into how the active and break phases of the South Asian monsoon influence subsurface chlorophyll in the Bay of Bengal, while also highlighting diel variability in bio-optical properties. Overall, the manuscript is well written and presents a comprehensive dataset of physical and bio-optical observations. I have provided several major and minor comments below aimed at clarifying methodological details, improving figure presentation, and strengthening the interpretation of the results.**

We thank the reviewer for their constructive feedback. We have carefully reviewed all comments and have improved the manuscript accordingly. We have copied the reviewer comments in bold and responded below them.

**The field campaign deployed three DBASIS systems. Do all three systems include the same set of bio-optical measurements? If so, it is unclear why the authors present only the ChlF data from M3. The methods section should be clarified to explain this choice. If all systems collected comparable measurements, it would be more informative to present averaged results or to justify focusing solely on M3. The methods section is difficult to follow and does not clearly convey these details.**

All floats observed a diel cycle in $ChlF_{SCM}$; however, the observations collected by M1 had some missing data due to compromised profiler performance. We considered showing the time series of all floats in the main section of the manuscript but ultimately decided additional time series did not enhance the manuscript or contribute to the results. We do, however, use the observations from all floats in Fig. 4 as this adds data points to our statistical analysis and makes it more robust. We then show the time series for M1 and M2 in Fig. S3 (supplementary).

We have justified our selection in the methods, as requested, "All three DBASIS floats had the same bio-optical instrumentation and generally collected comparable measurements. We present time series from only M3 here, which collected continuous (unlike M1) and high resolution (10 min) data, and show M1 and M2 observations in the supplementary materials. Our key results and findings were similar for all floats."

**The observations were conducted during the break and active phases of the southwest monsoon. Please provide the corresponding atmospheric conditions (just values) for these periods in the Results section, for example, wind speed, net heat flux, and cloud cover. This would help readers better understand the background atmospheric conditions associated with each phase.**

We have provided further information in the Results. As we did not directly measure cloud cover, we instead report on precipitation. We have added "Conditions at deployment were consistent with the southwest monsoon 'break' period with weak winds ($\sim$7.5 m s−1), large net heat fluxes into the ocean (QN , daily average QN of 180 W m−2), and no precipitation (Fig. 2)." The new Fig. 2 also shows wind speed and precipitation rate.

Then later in the Results, we add "From July 18 to 23, wind speeds decreased to an average of 4.4 m s−1, net heat fluxes were consistently large at a daily average of 139 W m−2, and the total rainfall measured 4.4 mm (Fig. 2). Subsequently, the atmospheric conditions transitioned from mostly sunny conditions to mostly cloudy conditions around July 23 with the onset of the 'active' phase of the southwest Monsoon (Fig. 2a). From July 23 to 28, surface-buoy-measured

shortwave radiation and net heat fluxes decreased, with a daily average QN of −39 W m−2, winds increased to an average of 8.8 m s−1, and rainfall remained low, totaling 28 mm (Fig. 2)."

**Chlorophyll concentration is primarily influenced by nutrient availability and light conditions. It is therefore unclear why the authors used an isopycnal criterion to identify the SCM instead of locating it directly from the maximum in the chlorophyll profile. The rationale for choosing the isopycnal based approach should be clarified and justified in the manuscript.**

We clarify that depth of maximum ChlF was identified first and then found to be within a narrow density range. We then proceeded to use a density criterion for the SCM, so we could analyze ChlF variability without internal wave heaving.

In Section 2.3 we have modified the text to "At all profilers, the depth of maximum ChlF was identified within a narrow density range. Excluding the initial 2 days observed at M1 (see supplementary materials), the SCM and the 1021.5 kg m−3 isopycnal were significantly correlated (r2 =0.85, p < 0.001) with a mean absolute error of 1.65 m. By defining the SCM as the depth of the 1021.5 kg m−3 isopycnal ±1 m, the SCM can be matched to an along-isopycnal reference frame. Given the internal wave fluctuations (∼15 m) observed during the campaign, employing an along-isopycnal reference frame was convenient for isolating the impact of changing surface irradiance and diffuse attenuation on ChlF at the SCM (denoted ChlF$_{SCM}$). The 1021.5 kg m−3 isopycnal had an average depth of 39 m at M3 (Fig. 2) that deepened to 50 m at M1."

**The authors calculate ITL depth as the depth where the temperature is ≥ 1°C from the surface value. What is the interpretation or treatment when the temperature difference is equal to 1°C?**

We have clarified the text, "Here, we define the ITL depth as the shallowest depth where the temperature was at least 1 ∘C cooler than the sea surface temperature (ΔT ≤1 ∘C), and the MLD as the shallowest depth where the density increase from the surface value corresponds to a temperature decrease of 1 ∘C."

**It is unclear why GCP is emphasized in this study. Since the field campaign directly measured bio-optical properties, the estimated GCP may carry considerable uncertainties. What would be the outcome if integrated ChlF were used instead of GCP? Does GCP provide additional information beyond what is captured by ChlF? The manuscript should include a justification for using GCP and clarify the added value it brings to the analysis.**

We have justified our use of GCP now in the methods and results, as well as highlight the limitations of using ChlF.

In the methods, we have clarified the benefits of analyzing GCP, "ChlF and integrated ChlF reflects the standing stock of chlorophyll, and therefore integrates pigment concentration over days to weeks. To understand the rate at which ChlF is produced and lost due to the local conditions at the SCM, we applied the diel cycle method (Nicholson et al., 2015; Barone et al., 2019). We identify the rate of gross ChlF production (GCP$_{SCM}$) and ChlF loss (code available at https://github.com/duebi/dielFit) to the NPQ corrected ChlF$_{SCM}$. This loss term describes all processes resulting in a reduction of ChlF$_{SCM}$, including grazing, mortality, and downward export."

And in the next paragraph, "The diel cycle method linearly fits three terms via ordinary least-squares fitting to $ChlF_{SCM}$. This allows us to directly evaluate the response of SCM production to intraseasonally modulated PAR, rather than relying on changes in the accumulated concentration."

In the results, we have added a new paragraph, "Our findings of light-limited ChlF could additionally be found by analyzing the instantaneous $ChlF_{SCM}$ and depth-integrated ChlF, but the relationships were much weaker. We compared depth-integrated ChlF over the upper 80 m to depth-integrated PAR, and $ChlF_{SCM}$ to $PAR_{SCM}$. Both linear regressions returned weak correlations ($r2 \leq 0.13$) that remained significant ($p < 0.001$, not shown). The instantaneous and depth-integrated ChlF was additionally influenced by ChlF loss and represents the accumulated biomass rather than just growth. Despite this, the strong influence of light availability on ChlF variability remained evident in both instantaneous $ChlF_{SCM}$ and $GCP_{SCM}$."

**The DBASIS array was deployed at the intersection of two eddies, but the presence of these eddies is not clearly visible in any of the figures. Additionally, the color bar used for chlorophyll is confusing and may hinder interpretation. Consider using a different color bar to improve clarity. Both chlorophyll and turbidity extend below 80 m depth on the initial date. Is this location near the coast? Please provide the actual depth at the measurement site. How do the authors determine that this signal is associated with a coastal plume rather than another source? Clarifying these points will help readers better interpret the observations.**

We have now split the original Fig. 1 into two new figures copied below. The new Fig. 1 plots satellite Chl, averaged from July 10 to 15. The coastal plume is shown and extends to the DBASIS profilers from the coastline to the west. Gray contours then show SSH, including the northern low SSH (<0.5 m) eddy and southern high SSH (>0.5 m) eddy. The precise SSH values are more obvious in the smaller zoomed-in panel. The colorbar for ChlF is now more obvious, and we adjusted the colors so that the lighter colors were removed. The figure caption now states the depth and distance from coast of the profilers, which are around 2,600 m deep (±200 m due to movement of float) and 400 km from the coast, respectively.

The new Fig. 2 then shows the time series data that was originally shown in Fig. 1, plus new panels showing wind speed, precipitation rate, and subsurface PAR.

In the Results, we have added "The DBASIS array was deployed at the intersection of two eddies, as inferred from sea surface height (SSH, Fig. 1). Low-SSH (SSH < 0.5 m) and high-SSH (SSH > 0.5 m) eddies were present to the northwest and southwest, respectively. Between these two eddies, a plume of elevated surface chlorophyll was advected more than 400 km from the coast toward the DBASIS array (Fig. 1a). This coastal plume exhibited elevated turbidity and surface chlorophyll-a fluorescence (ChlF), both of which reduced light penetration into the upper ocean (Fig. 2). Similar coastal plumes have been observed elsewhere at the intersection of cyclonic and anticyclonic eddies (i.e., eddy dipole, Malan et al., 2020)."

[Figure]

*Figure a: Overview of observations. a) Bay of Bengal with the study region (red square), location of analyzed biogeochemical (BGC) Argo profilers deployed from 2013 to 2018 (yellow, WMO ID: 2902086, 2902087, 2902114, 2902158, 2902160, 2902161, 2902189, 2902196, 2902217, 2902264), with a further analyzed 2016 profile highlighted (pink, WMO ID:2902193, Argo, 2000), location of DBASIS floats (purple), 5-day averaged (10-15 July, 2019) satellite chlorophyll (Chl, background color; Sathyendranath et al., 2019, 2023), and gray contours of the EU Copernicus Marine Service global ocean 1/12° sea surface height (SSH, 4th July, 2019; https://doi.org/10.48670/moi-00021). b) Position and date for the three high-resolution DBASIS floats: M1 (dark blue), M2 (light blue), and M3 (green), with the depth-maximum chlorophyll-a fluorescence (ChlF) shown for M3. Gray contours show SSH (m). DBASIS floats were located in depths ~2600 m deep and ~400 km southeast of the nearest coastline.*

[Figure]

*Figure b: Observations at M3. a) the net heat flux (QN ), b) wind speed (blue left y-axis, Ws) and precipitation rate (orange right y-axis, P ), c) Photosynthetically available radiation (PAR), d) ChlF, and e) turbidity (τ ). In c) to e), two isopycnals (1021.5 kg m−3 and 1023 kg m−3) are contoured (black), and in d) the depth of the isothermal layer (ITL, pink) and mixed layer depth (MLD, yellow) are shown.*

**At the SCM, ChlF was initially relatively low and then doubled after the drifting platform moved away from the plume. This reflects the ChlF conditions between two distinct regions. Please highlight this observation in the manuscript, as it helps illustrate the spatial variability in phytoplankton distribution. Since the observations cover two distinct regions - one influenced by a coastal plume and one without, it would be helpful to include a satellite chlorophyll-a map with bathymetry contours and the observation locations overlaid. This would allow readers to easily visualize and understand the spatial distinction between the plume and non-plume regions.**

In the Results, we have added "Over the 19-day deployment, the array drifted southeastward and sampled two distinct regimes– one influenced by a coastal plume and one outside its influence."

The new Fig. 1 also plots satellite Chl, as requested. We attempted to add bathymetry contours; however, the figure was too cluttered and so instead state the seabed depth in the Fig. 1 caption. The difference between the plume and non-plume regions are now shown in Figs. 1 and 2.

**On July 17, low-salinity water was observed within the top 10 m, after which the MLD became shallower. Could this be due to a rain-induced surface layer? Is it plausible for rainfall to generate a low-salinity layer with 10 m thickness?**

Yes, as stated in section 3.3, "A passing storm resulted in a shallow rain-layer forming around July 17, the rapid shoaling of the MLD, diverging surface and MLD temperature and salinity (see Kerhalkar et al., 2025), and a rapid decrease in ChlF at the MLD base (Fig. 5)." We describe only briefly here as it is not the focus of the manuscript, but we observed a rain-induced surface layer from the July 17 rainfall. Salinity was decreased to a depth of 5 m, which is more obvious in Kerhalkar et al., (2025). We now reference this manuscript.

**The manuscript mentions a diel cycle of the derived gross production and loss terms in the Discussion, but this is not clearly shown in the Results section. Please include evidence or plots in the Results to support this point. Additionally, it is unclear how the authors conclude that the SCM represents a maximum in biomass—this needs to be clearly demonstrated in the Results section.**

Thank you for highlighting this issue. We now clarify, "During the 2019 monsoon in the central Bay of Bengal and away from the coastal plume, ChlF and turbidity had a subsurface maximum, denoted the subsurface chlorophyll maximum (SCM, Fig. 2), and the SCM almost always coincided with the 1021.5 kg m−3 isopycnal. Both ChlF and turbidity also had a diel periodicity at the SCM, further suggesting that variations in ChlF at the SCM represent similar changes in biomass, as has been previously shown in the Indian Ocean waters (Cornec et al., 2021; Prasanth et al., 2023). The diel periodicity in ChlF has been observed elsewhere in the BoB (Lucas et al., 2016; Prasanth et al., 2023), and highlights a tight coupling between phytoplankton growth and loss in these oligotrophic waters. At the SCM, the gross production derived from both ChlF and turbidity varied with $PAR_{SCM}$ (Fig. 4)."

Following the above comment regarding the SCM, we now validate our definition of the SCM in the Methods section 2.2. In the results section 3.1, we have added, "ChlF was consistently enhanced near the subsurface chlorophyll maximum (SCM)." The maximum in ChlF at the SCM is also evident in Figs. 2d, 3a, 5c, and 6b.

In the Results, we have further explained, "ChlF at the subsurface chlorophyll maximum (SCM, $ChlF_{SCM}$) was initially relatively low and then approximately doubled after the drifting platforms left the plume (July 14 to 20). Meanwhile, turbidity near the SCM ($\tau_{SCM}$) initially decreased and subsequently increased as $ChlF_{SCM}$ increased (Fig. 2d-e). Outside the plume, turbidity is expected to primarily reflect algal biomass. The observed turbidity outside the plume, especially with its co-variability with ChlF near the SCM, indicates that variability in ChlF reflects changes in biomass and not just pigment concentrations."

Then later in section 3.2 we state "The estimated gross turbidity production at the SCM ($GTP_{SCM}$) linearly varied with PAR at the surface and SCM, similar to $GCP_{SCM}$ variability (Fig. 4b,d). This indicates the co-variability of $ChlF_{SCM}$ and light availability is not limited to pigment concentration (e.g., photo-acclimation) but also corresponds to changes in biomass."

**Figure 7b shows clear diel variability, and I am curious how the authors identified vertical migration of zooplankton from this figure. Providing a brief explanation in the manuscript would be very useful for readers who are not familiar with this type of observation and help interpret the biological patterns more clearly.**

Thank you for your feedback. We have elaborated on how Fig. 7 (now Fig. 8) shows diel vertical migration in Section 4.2, "The backscatter measurements showed enhanced values at night in shallow waters with depths less than 50 m and density of 1022 kg m−3 or less (Fig. 8). Then during the day, backscatter measurements were enhanced in deeper waters at around 60 m

depth. These observations are consistent with a shallow diel migration of zooplankton between around 60 m at daytime to near the SCM at nighttime."

**Use ChlF$_{SCM}$ instead of SCM ChlF, similarly SCM PAR as PAR$_{SCM}$.**

We have changed our notation throughout, and for consistency, we changed GCP to GCP$_{SCM}$.

**What is phase-locking of diel cycle with irradiance?**

In the Introduction, we now state, "The tight coupling between the diel cycle and the solar cycle (i.e., sunrise and sunset) …".

Then in Section 3.2, we now clarify, "The consistent timing of irradiance and the diel cycle (e.g., maximum in ChlF rate of change at noon, not shown) has been used to estimate gross production and loss …".

**Figure 4c is a repetition of Figure 1d. Add ML, ILT to Figure 1d.**

Added to Fig. 2.

**Why does Figure 2b start at 20 m depth, whereas Figure 1d shows data from the surface? Additionally, Figure 2b shows some black areas at the top—please clarify what these represent. Consistency or explanation of the depth ranges would help readers interpret the figures more clearly.**

In Section 3.2, we now clarify "The rapid vertical profiling of the Wirewalkers allowed the assessment of bio-optical variability from a time-varying, along-isopycnal frame of reference (Fig. 2d vs. 3a). We use the observed density to convert ChlF to an along-isopycnal reference frame. This approach moderates the effect of passing internal waves that confound measurements taken at discrete depths and highlighted the variability of ChlF at diel timescales (Fig. 3). Due to high variability in surface densities, we omit the upper 20 m in the along-isopycnal figure (Fig. 3a). We also note that due to the relatively dense surface densities from July 12 to 18, the along-isopycnal figure has missing data (black regions) to up to 30 m depth (≤1021.3 kg m−3 isopycnal), as no ChlF observations were made at these densities."

**Check for typos in the manuscript, for example - 40% percent ("percent" is not required here)**

Thank you for identifying this typo. We have corrected here and other identified errors.

**Figure 1b has some unwanted axis labels (30)**

Figure has now been replaced.

**I could not find the time derivatives of chlorophyll in the manuscript.**

We no longer refer to the time derivatives of ChlF, and instead replaced the text with, "The diel cycle at the SCM showed peak concentrations around 3 hours after local noon and minimum concentrations at dawn (Fig. 3a,c)."

**After the buoys moved away from the plume on the fourth day, surface PAR was also high, which may have contributed to the increased PAR at the SCM. It would be helpful if the authors could provide vertical profiles of PAR for each condition to better illustrate the light environment influencing the SCM.**

We have added subsurface PAR to the new Fig. 2. We note the impact of the plume is also evident in Fig. 3b, where the surface (purple) and SCM (teal) line plots diverge the most within the coastal plume.

**Where was the GCP calculated - at SCM specifically, or over the entire water column? Please clarify this in the Methods or Results section to avoid confusion.**

We have clarified throughout by replacing GCP with GCP$_{SCM}$, as well as clarifying our text throughout the manuscript.

**Add y-axis labels on Figures. For example Figure 4 d to f.**

Thank you for identifying the missing labels. We have fixed Figure 4 (now Fig. 5). All other figures have appropriately labelled axes.

**Please specify the height at which the buoy's meteorological package measured the atmospheric parameters.**

We have added the instrument heights to the Methods, "The buoy's meteorological package measured wind (2.9 m above mean sea level; MSL), air temperature (2.6 m above MSL), humidity (2.6 m above MSL), precipitation (2.9 m above MSL), barometric pressure (2.3 m above MSL), sea surface temperature (0.3 m and 1.5 m below MSL), and downwelling solar and infrared radiation (3.2 m above MSL)."